# Drug Discovery for Cutaneous Leishmaniasis: A Review of Developments in the Past 15 Years

**DOI:** 10.3390/microorganisms11122845

**Published:** 2023-11-23

**Authors:** Hannah N. Corman, Case W. McNamara, Malina A. Bakowski

**Affiliations:** Calibr at Scripps Research, La Jolla, CA 92037, USA; cmcnamara@scripps.edu (C.W.M.); mbakowski@scripps.edu (M.A.B.)

**Keywords:** *Leishmania*, cutaneous leishmaniasis, drug discovery

## Abstract

Leishmaniasis is a group of vector-borne, parasitic diseases caused by over 20 species of the protozoan *Leishmania* spp. The three major disease classifications, cutaneous, visceral, and mucocutaneous, have a range of clinical manifestations from self-healing skin lesions to hepatosplenomegaly and mucosal membrane damage to fatality. As a neglected tropical disease, leishmaniasis represents a major international health challenge, with nearly 350 million people living at risk of infection a year. The current chemotherapeutics used to treat leishmaniasis have harsh side effects, prolonged and costly treatment regimens, as well as emerging drug resistance, and are predominantly used for the treatment of visceral leishmaniasis. There is an undeniable need for the identification and development of novel chemotherapeutics targeting cutaneous leishmaniasis (CL), largely ignored by concerted drug development efforts. CL is mostly non-lethal and the most common presentation of this disease, with nearly 1 million new cases reported annually. Recognizing this unaddressed need, substantial yet fragmented progress in early drug discovery efforts for CL has occurred in the past 15 years and was outlined in this review. However, further work needs to be carried out to advance early discovery candidates towards the clinic. Importantly, there is a paucity of investment in the translation and development of therapies for CL, limiting the emergence of viable solutions to deal with this serious and complex international health problem.

## 1. Introduction

Leishmaniasis is a vector-borne disease caused by protozoan parasites of the genus *Leishmania*. Female phlebotomine sandflies, of the genus *Phlebotomus* in the Old World and the genus *Lutzomyia* in the New World, transmit the parasite during blood feeding. There are 50 proven species of sandfly that can transmit disease to humans, while the estimated number of species capable of transmission is closer to 80 [1]. Leishmaniasis is endemic in nearly 100 countries, with approximately 350 million people at risk of infection. The overall prevalence of leishmaniasis has been estimated to be 12 million cases, with 2 million new cases occurring annually; however, these incidence numbers are most likely underestimated due to no mandatory reporting. Importantly, the World Health Organization (WHO) has categorized leishmaniasis as one of 20 neglected tropical diseases (NTDs) [2], emphasizing the significant impact this disease has around the world and the lack of effective treatment options.

Leishmaniasis has a spectrum of clinical presentations depending on the infecting species. Visceral leishmaniasis (VL) is typically caused by *L. donovani* and *L. infantum*. Its characteristic symptoms include persistent irregular fever, splenomegaly, and hepatomegaly [3,4,5,6]. Cutaneous leishmaniasis (CL) is caused by over 20 species, with the majority of cases caused by *L. tropica*, *L. aethiopica*, *L. major*, *L. infantum*, *L. mexicana*, *L. amazonensis*, *L. braziliensis*, and *L. guyanensis* [7,8]. While considered a mild disease manifestation, substantial cosmetic morbidity, social stigmatization, and other psychological burdens greatly impact those affected [9,10]. CL is characterized by a lesion that develops at the site of a sandfly bite; lesions can develop as a papule, which then enlarges to a nodule that then ulcerates. Mucocutaneous leishmaniasis (MCL) is a devastating, disfiguring disease that develops in between 1 percent and 10 percent of CL cases [7,11,12]. MCL is characterized by destructive lesions of the nasopharyngeal mucosa and cartilage in the nasal septum and palate.

The clinical presentation, or lack thereof, of leishmaniasis is influenced by host genetics, species of parasites, and the nature of the immune response invoked. For example, Old World species, like *L. major* and *L. infantum*, tend to cause self-limiting ulcers, while New World species, such as *L. amazonensis* and *L. braziliensis*, can be severely destructive, especially with MCL [13]. After a sandfly infection, a nodule progresses to an ulcerated lesion that will heal over months if the patient is immunocompetent. These nodules tend to be painless, but as lesions further develop and become ulcerated, they are susceptible to secondary infections with bacteria or fungi. Symptom onset can be quite variable, ranging from days to years post sandfly bite; the average time ranges from 2 weeks to 8 weeks [14,15].

CL is the most prevalent clinical manifestation of leishmaniasis, with a range from 600,000 to 1 million new cases reported annually [13]. Importantly, ninety percent of these cases are reported from eight countries, namely Afghanistan, Algeria, Brazil, Iran, Pakistan, Peru, Saudi Arabia, and Syria [7]. High population densities and rampant malnutrition combined with poor sanitation facilities are common among CL endemic areas; human migration, climate change, political instability, and warfare also contribute to expanding endemic regions and increasing the propensity for epidemics worldwide [16,17,18,19].

Chemotherapy has been the primary focus in the battle against leishmaniasis due to difficulties with vector control [20,21] as well as limited success with vaccine development [22,23,24,25]. Female phlebotomine sandflies can be indoor feeders (endophagic) or outdoor feeders (exophagic) [21]. While research has shown that sandflies are highly sensitive to insecticides, resistance to a common insecticide DDT has been reported [26]. Vector control methods using insecticides, like indoor residual spraying (IRS) or treated bednets (ITNs), are widely used to control endophagic flies, but they need to be repeated regularly [27] and do not address the exophagic fly population. Insect repellant or protective clothing are preventative measures for exophagic flies [20] but may not be practical or affordable to socioeconomically disadvantaged populations in endemic areas [28]. Treatment plans and chemotherapies for leishmaniasis have generally been prioritized for VL, as CL tends to be self-limiting and self-healing in immunocompetent patients. Increased use of chemotherapies for CL have been occurring due to comorbidities including HIV, the possibility of disfiguring scars, nodular lymphangitis, and lasting disability from permanent destruction of tissues [13]. The current drugs used are decades old, with several limitations including toxicity, severe side effects, excessive cost, long treatment regimens, and emerging drug resistance. Pentavalent antimonials are often considered the first-line treatment option in many endemic areas. Miltefosine, pentamidine isethionate, amphotericin B, and paromomycin are now being used either in combination with antimonials or as the primary treatment. Even with multiple compounds available, many CL patients will not receive treatment; the decision to treat is multifaceted and often driven by a need to accelerate a cure if lesions are present for more than 6 months [7], are located in a sensitive area like the face [7,29,30], and often to reduce the risk of dissemination or progression to MCL [7]. In endemic areas, medical experts prefer localized therapies when less than five lesions are present, but will utilize systemic treatment options for multiple lesions, facial involvement, or when topical treatment is not feasible [13,29]. If a patient is infected with a species known to cause MCL (like *L. braziliensis*), systemic treatment is typically favored along with combination therapies and close monitoring.

With current treatment limitations, especially for CL, an emphasis has been placed on the identification and optimization of new chemical entities suitable for leishmaniasis treatment, with target product profiles established by the DND*i* (Table 1). 

A major problem for any potential anti-leishmanial compound is their transport across multiple host cell membranes as well as stability within the acidic parasitophorous vacuole of the macrophage, where the parasites reside and replicate (Figure 1).

While researchers have identified novel compounds and treatment regimens for VL, promising therapeutic breakthroughs for CL are decades away from use in the clinic [13,32]. To address these concerns, researchers have been utilizing both phenotypic and target-based compound screening and development. Phenotypic screens are an effective approach to select compounds that impact overall parasite viability; these are especially helpful, as there are limited *Leishmania*-specific targets that have been identified and validated to be druggable candidates [33]. Additional benefits of utilizing phenotypic screens include determination of potential off-target host cell toxicities, and insights into compound permeability and stability within the host–parasite microenvironment [33]. However, deconvolution to determine a specific molecular target or mechanism of action for a compound is a major challenge. In the previous two decades, there have been several druggable targets identified for *Leishmania* parasites, including glucose 6-phosphate isomerase [34], triosephosphate isomerase [34], trypanothione reductase and synthetase [35], metacaspses [36], aspartic protease [37], kinetoplastid topoisomerase II [38], dihydrofolate reductase [39], disulfide isomerase [40], and the proteasome [41,42,43,44,45].

In this review, drug discovery efforts focused on CL from the previous 15 years were discussed. We performed a literature search utilizing PubMed to identify published research between the years 2000 and 2023 focused on the prominent species causing CL, namely *L. major*, *L. tropica*, *L. amazonensis*, *L. braziliensis*, and *L. mexicana*. We filtered results using the MESH term ‘drug discovery’, as well as excluded review articles to prioritize primary research articles. We have focused on the current developments in treatment options as well as potential molecular targets, with results ranging from *Leishmania*-specific to pan-kingdom research. 

## 2. Drug Discovery Targeting a Single *Leishmania* Species

Many researchers have focused their drug discovery efforts on a single species of *Leishmania* due to practical considerations and in hopes of increased specificity with limited adverse effects. Techniques like Cos-Seq [46], quantitative real-time PCR [47,48], quantitative proteomics [49], metabolic network analysis [50], homology modeling and docking [51,52], and advanced optical imaging technology [53] have allowed researchers to identify several targets and compound classes effective against *L. major*, *L. infantum*, *L. tropica*, *L. braziliensis*, and *L. mexicana*. Lackovic et al. (2010) identified several compounds that target *L. major* GDP-mannose pyrophosphorylase [54]. While these compounds were relatively effective against the enzyme itself, with IC_50_ values less than 10 µM, they were substantially less effective against the amastigote life stage of the parasite, with IC_50_ values increasing by 2–10-fold. Similarly, researchers have identified inhibitors of *L. mexicana* cysteine protease B utilizing purified enzymes and molecular docking studies [55,56,57], but these results were not confirmed with either promastigotes or amastigotes. Using purified *L. mexicana* GAPDH, researchers employed computer modeling to identify potential inhibitors and conformational dynamics [58,59]; however, these findings were not confirmed in vitro with purified enzymes, promastigotes, or amastigotes. Inhibitors of *L. braziliensis* histone deacetylase were identified in high-throughput screens against promastigotes, with activity confirmed against intracellular amastigotes [60]. While several compounds were not potent against promastigotes, with IC_50_ values ranging from 27 µM to greater than 80 µM, they exhibited IC_50_ values less than 10 µM in the intracellular infection model. These results are exciting, but additional research to validate these molecular targets is still needed prior to clinical development.

In addition to molecular targets, researchers have focused on natural products from plants, fungi, and bacteria that were found to exhibit potent activity against *L. tropica* [61,62], *L. mexicana* [63,64], and *L. braziliensis* [65,66]. Awada et al. (2022) first isolated bacterial samples from Lebanon and subsequently extracted bacterial metabolites [61]. These metabolites were evaluated for anti-promastigote activity as well as potency in an intramacrophage infection model; importantly, HAS1, isolated from soil *Streptomyces*, significantly inhibited *L. tropica* amastigote replication in THP-1 macrophages. Similarly, Rodrigues et al. (2019) screened cinnamic acid derivatives against promastigotes and intramacrophage amastigotes, with most compounds screened more active against amastigotes [66]. Mbekeani et al. (2019) harvested fermentation products from fungal cultures to screen against *L. mexicana* amastigotes [64]. Interestingly, the preliminary screening was completed using axenic amastigotes, while activity was confirmed in an intramacrophage infection model; while axenic amastigotes provide researchers the opportunity to utilize a more clinically relevant life stage of the parasite without the complications of a host cell, these results do not necessarily translate to the intramacrophage model.

A substantial amount of research has been focused on *Leishmania amazonensis*, a predominant species in the New World with potential for chronic disease [67]. Researchers have developed and validated several bioluminescent reports for use in vitro [68] and in vivo [69,70]. With these tools, researchers have identified natural products and synthetic compounds with potent anti-leishmanial activity as well as potential drug targets. Several examples of these potential targets include arginase [71,72,73] and sterol biosynthesis [74]. Researchers first screened compounds against purified *L. amazonensis* arginase, with activity confirmed using promastigotes and intramacrophage amastigotes [71,73]. Similarly, Andrade-Neto et al. (2016) documented morphological differences between untreated and treated promastigotes, with additional characterization of the overall sterol composition of the treated promastigotes, and finally confirmed activity using intramacrophage infection rates [74]. Other mechanisms of action that have been identified include autophagy and the production of reactive oxygen species, resulting in decreased macrophage infection rates in as little as 3 hours of treatment with apigenin [75] and altering the interaction between macrophage membranes and the promastigote, resulting in decreased macrophage infection [76]. These in vitro results are promising, but no in vivo work has been conducted thus far to confirm their potency or safety.

In addition to synthetic compounds, researchers have utilized bacterial- [77] and fungal- [78] produced bioactive derivatives that alter *Leishmania amazonensis* mitochondrial function in both promastigotes and amastigotes with limited mammalian toxicity [77]. Natural products derived from plant sources have also exhibited anti-leishmanial activity, including sesquiterpene lactone-rich fractions from *Tanacetum parthenium* (L.) Schultz-Bip [79], fruit juice from *Morinda citrifolia* Linn. [80], and essential oils from *Artemisia absinthium* [81]. While Almeida-Souza et al. (2018) only demonstrated potential anti-leishmanial activity against axenic amastigotes [80], Monzote et al. (2014) identified essential oils that inhibited promastigote and amastigote growth in vitro, as well as reduced lesion size and parasite burden in vivo [81]. Similarly, Rabito et al. (2014) first determined the anti-proliferative activity of the sesquiterpene lactone-rich fractions against promastigotes and axenic amastigotes, with no significant difference in their IC_50_ values [79]. Then, they treated *L. amazonensis*-infected BALB/c mice with the same fraction using intramuscular injections once every 3 days; importantly, there was a significant reduction in parasite load compared to the untreated mice, as well as little to no adverse effects.

Many compound classes have exhibited activity in vitro against promastigotes or amastigotes, including thiophene-indole hybrids [82,83], 2-amino-thiophene derivatives [84], pyrazolo-thiophene hybrids [85], camphor hydrazone derivatives [86], tetroxanes [87], isoxazole derivatives and tetrahydrofuran neolignans [88], piperine derivatives [89], naphthotriazolyl-4-oxoquinolines [90], furazolidone-based cyclodextrins [91], 2-pyrimidinyl-hydrazone and N-acylhydrazone derivatives [92], piperidine-benzodioxole derivatives [93], and methyl gallate [94]. In vivo studies of these compounds are required to elucidate their activity and safety profiles before clinical development can begin. Other compound classes that have been demonstrated to be potent in vitro and in vivo include 4-nitrophenylacetyl and 4-nitro-1H-imidazolyl [95], and N,N′,N″-trisubstituted guanidines [96].

Interestingly, with so many new and diverse chemical compounds with potent activity, researchers have also employed combination therapies with existing and preapproved compounds. The selective estrogen receptor modulator tamoxifen has previously been found to be effective against *L. amazonensis* in vivo, reducing lesion size as well as parasite burden in the BALB/c model [97]. Trinconi et al. (2014) expanded on these results by combining tamoxifen and amphotericin B both in vitro and in vivo [98]. Potential compound associations were evaluated using the fixed ratio isobologram method [99,100] and odds analysis [101], and the results revealed an indifferent interaction in vitro. However, in vivo analyses revealed that even low-dose combinations of tamoxifen and amphotericin B reduced lesion size and parasite burden significantly compared to untreated animals and either tamoxifen or amphotericin B alone [98]. By utilizing preapproved compounds, this treatment combination may be able to be employed in clinics much sooner than more novel compounds.

## 3. Drug Discovery Targeting Multiple Species of *Leishmania*

CL can be caused by multiple species of *Leishmania*, and similar clinical presentations and overlapping endemicity make developing a broad anti-*Leishmania* therapy a logical choice. However, this is often easier said than done; high genetic and phenotypic variability between species and strains within a single species are incredibly difficult barriers. For example, Alcantara et al. (2020) conducted a multi-species intramacrophage phenotypic screening assay using the commercially available compound library LOPAC [102]. Importantly, 51 compounds of the total 1280 were considered active, and of those 51 active compounds, only 14 presented broad-spectrum activity, resulting in a pan-active hit rate of 1.09% [102]. 

Researchers have established several tools useful for broad-spectrum anti-leishmanial drug discovery, including parasites constitutively expressing fluorescent proteins [103,104], differential protein expression of different parasite life stages [49], automated image analysis protocols [105], proteome mining [106], and kinome mining [107]. These tools were useful in identifying several potential drug targets, including inositol phosphorylceramide synthase using purified *Leishmania* enzyme in a plate-based assay [108], serine proteases through activity-based protein profiling of promastigotes [109], the Lmj_04_BRCT protein domain first characterized via homology modeling and then validated in vitro using intracellular amastigotes [110], cysteine protease CPB2.8(Δ)CTE via enzymatic screening followed by intramacrophage screening [111], and the Hsp90 chaperone of promastigote parasites [112].

An early natural product screened for activity against *Leishmania* was dillapiole, a compound isolated from *Piper aduncum.* Dillapiole, along with two phenylpropanoid derivatives, were evaluated against *L. amazonensis* and *L. braziliensis* [113]. Interestingly, dillapiole itself was found to be the most potent structure, but the IC_50_ values against *L. amazonensis* and *L. braziliensis* were 69.3 µM and 59.4 µM, respectively, with substantial cytotoxic effects on fibroblast cells, while the derivatives were significantly less potent (with less cytotoxicity noted) [113]. More recently, Oliveira et al. (2021) identified isopentenyl caffeate as a more promising anti-leishmanial compound, with in vitro IC_50_ values against promastigotes and amastigotes of *L. amazonensis* and *L. chagasi* under 2 µM, with selectivity indices over 100 [114]. Similarly, Van Bocxlaer et al. (2019) identified three nitroimidazoles, one benzoxaborole, and three aminopyrazoles that exhibited potent activity across several *Leishmania* species, with their in vitro IC_50_ values ranging from 0.29 µM to 18.3 µM [115]. Most importantly, these selected compounds had high levels of efficacy in a murine CL model, with significant reductions in lesion size as well as a 2-log-fold reduction in parasite load compared to the untreated control, exhibiting excellent activity with limited adverse effects. Thiazolopyrimidine derivatives were designed by Istanbullu et al. (2020) to target *L. major* pteridine reductase 1 (LmPTR1); these researchers were able to identify one potent compound with potent activity against the purified enzyme, as well as IC_50_ values of 7.5 µM and 2.69 µM against promastigote and intracellular amastigotes, respectively [116]. In addition to those mentioned above, other compound classes with activity against multiple CL-causing species were identified, including chalcone-like hybrids [117], monovalent ionophores [118], benzimidazole derivatives [119], and cruzioseptins [120]. 

Since VL can be fatal, many researchers have tried to identify compounds that would be effective against multiple species, causing multiple clinical presentations of leishmaniasis. High-throughput screening has been employed by several researchers either as phenotypic screens [32,102,121,122] or target-specific, such as *Leishmania* protein disulfide isomerase [40,123]. Several compound classes have been identified with potent anti-leishmanial activity, including substituted 1,2-dioxanes [124], mono-arylimidamides [125], pterocarpanquinones [126], C-10b-substituted dihydropyrrolo [1,2-b]isoquinolines [127], amino-substituted 1H-phenalen-1-ones [128], and chalcones [129]. For example, Ortalli et al. (2018) synthesized 31 novel chalcone compounds, with 16 compounds showing activity against *L. donovani* promastigotes [129]. Of those sixteen compounds, two showed greater than 50% inhibitory activity in the intramacrophage model. Interestingly, one of these potent compounds interacted with *Leishmania* trypanothione reductase with high affinity and sub-micromolar potency. These in vitro results, while interesting, must be confirmed in an in vivo model prior to any hopes of clinical relevance.

In addition to in vitro screening assays, computational analyses have also allowed researchers to identify and optimize potential broad-acting chemotherapies. For example, Collar et al. (2011) first screened 55 arylimidamides against *L. donovani* axenic amastigotes and *L. amazonensis* intracellular amastigotes, and then utilized three-dimensional QSAR modeling and GALAHAD modules to identify important structural components of active compounds [130]. Researchers have also screened compounds against specific *Leishmania* targets, including the cytochrome bc1 complex [131], and cysteine protease CPB2.8(Δ)CTE [132]. Biochemical analyses have also been employed to identify potential *Leishmania* targets, including nucleoside diphosphatase kinase (NDK) [133], GDP-mannose pyrophosphorylases [134], and *Leishmania* N-myristoyltransferase [135]. 

The *Leishmania* proteasome, the primary cellular protease, and subsequent inhibitors have been investigated for decades. In addition to the classical eukaryotic 20S and 26S proteasomes, *Leishmania* parasites possess a bacterial-like protease complex called HsIVU [42,44], offering additional potential targets. The *L. mexicana* 20S proteasome was shown to be sensitive to a proteasome-specific inhibitor; however, that same inhibitor was found to be inefficient at inhibiting parasite growth in vitro [136]. Interestingly, two known proteasome inhibitors used to treat HIV exhibited a dose-dependent and irreversible growth inhibition on *L. major* and *L. infantum* promastigotes [137], suggesting a potential use when patients are coinfected with *Leishmania* and HIV. Another well-known protease inhibitor, lactacystin, also exhibited a dose-dependent growth inhibition of *L. chagasi* [138]. Silva-Jardim et al. (2004) also found that treatment with lactacystin greatly affected parasite survival inside host macrophages; after 96 hours, only 2% of treated parasites were viable compared to 81% of non-treated parasites [138]. More recently, a selective 20S proteasome inhibitor was identified and optimized for treatment of multiple kinetoplastid diseases. The inhibitor GNF6702 contains a triazolopyrimidine scaffold that inhibits chymotrypsin protease activity without hindering mammalian proteasomes [41,43]. A structurally similar protease inhibitor, LXE408, is in clinical trials for visceral leishmaniasis [45,139]. This inhibitor is promising due to its high efficacy in mouse models as well as its relative safety and tolerability with oral administration [45]. These clinical candidates have been tested against several species that cause VL, but limited work has been carried out with CL-causing species. 

## 4. Drug Discovery Targeting Multiple Eukaryotic and Prokaryotic Pathogens

Since kinetoplastid parasites like *Leishmania* spp. and *Trypanosoma* spp. are closely related, researchers are currently identifying compounds and potential molecular targets with activity against both genera. Novel inhibitors of trypanothione synthetase [140] and phosphoglucose isomerase [141] were identified using high-throughput screening techniques and molecular docking studies. Both groups found substantial structural similarities between genera [140] and suggestions of similar compound-binding pockets [141]. In addition to common targets, researchers have identified several compound classes and sources that are effective against kinetoplastids. Compounds like 3,5-disubstituted isoxazoles have been found to be active against *T. cruzi* [142,143], *L. amazonensis* [142,143], and *L. braziliensis* [143]. Similarly, prenylated chalcones [144] and prenyloxy chalcones [145] have demonstrated both anti-leishmanial and anti-trypanosomal activity. Additional compound classes that have shown potent anti-kinetoplastid activity include binuclear cyclopalladated compounds [146], 1,3,4,5-tetrasubstituted pyrazoles [147], N,N′-dihetaryl substituted diamines [148], N-benzene and N-naphthalenesulfonamide derivatives [149], thiazolyl-isatin derivatives [150], and ruthenium–purine complexes [151]. Natural products have also been an area of interest, with several researchers identifying compounds like sesquiterpene lactones [152,153] and terpenoids [154] with anti-trypanosomatid properties.

While researchers are identifying effective compounds and potential targets for multiple genera of kinetoplastid parasites, additional research is being carried out more broadly on targeting protozoan parasites in general. As mentioned above, chalcones are known anti-trypanosomatid compounds, and recent research has shown that chalcones exhibit broad anti-parasitic activity against *Plasmodium* parasites as well [155]. Known compounds like clinically approved HDAC inhibitors [156] and those found in the Medicines for Malaria Venture pathogen box [157] have shown substantial activity against *Leishmania*, *Plasmodium*, and even *Schistosoma* parasites. In addition, sesquiterpene glycosides [158], flavonoids [159], quinoxaline derivatives [160], and quinazoline derivatives [161] were shown to possess efficacy against *Plasmodium* and kinetoplastids. Interestingly, researchers have even identified compounds effective against protozoan parasites, including *Leishmania*, *Trypanosoma*, *Giardia*, and *Trichomonas* [162,163]. 

Researchers have also begun looking for broadly active anti-infective agents that would have activity against protozoans, bacteria, and even fungi. This is a logical next step, considering that bacteria and fungi are often opportunistic pathogens that can coinfect a CL lesion. In addition, amphotericin B is a known anti-fungal agent with potent anti-leishmanial properties. Researchers have isolated essential oils from traditional medicines in CL endemic areas that have potent anti-leishmanial and anti-dermatophyte activity [164,165,166], and even anti-bacterial activity [167]. The 3,5-disubstituted isoxazole compound class has been shown to be potent against protozoan parasites as well as opportunistic *Candida* species [168], while 1,2,4-triazole clubbed Mannich bases have been shown to be potent against protozoans, *C. albicans*, and *M. tuberculosis* [169]. Researchers have identified bioactive peptides targeting phospholipase A2 that are effective against *Leishmania*, *S. aureus*, and *E. coli* [170]. Interestingly, natural products isolated from *S. phalerata* essential oils had potent activity against *Leishmania*, Gram-positive, Gram-negative, aerobic, and anaerobic bacteria [171]. *Mycobacterium tuberculosis*, the causative agent of tuberculosis, can often present a comorbidity with leishmaniasis. Researchers have found several compound classes that are effective against bacteria and parasites, including naphthoquinone hybrids [172], deaminated terpenoids [173], hybrid furoxanyl N-acylhydrazone derivatives [174], and N-{2-[(7-chloroquinolin-4-yl)amino]ethyl}ureas [175]. 

## 5. Conclusions

With the plethora of research into identifying safe and efficacious anti-leishmanial compounds, several questions arise: (1) Is there a benefit to multi-target approaches active against several different organisms? (2) Would researchers’ efforts be better spent focusing on compounds that are effective against multiple species of *Leishmania*? (3) Is single-target drug discovery the most beneficial to the need for chemotherapies at this time? (4) How can the large body of early discovery work be effectively translated to the clinic? 

Traditional drug discovery has been described as “on-target”, meaning compounds are designed to target a single molecular entity with high selectivity. This high selectivity ideally prevents unwanted effects due to compound interactions with other molecular targets. For anti-leishmanial research, this can be a difficult task due to the serious impediments and knowledge gaps within *Leishmania* biology itself. There are a limited number of fully validated targets and limited ways to confirm on-target effects of active compounds [176]. In addition, *Leishmania* parasites exhibit a stochastic aneuploidy in their genomes; chromosome polyploidy has been cited in *L. major* chromosome 31 [177] and *L. donovani* chromosome 15 [178]. Importantly, this malleability of the genome often hinders the basic reverse genetic approaches that researchers typically use for phenotyping and compound target identification. High selectivity has been a top priority in anti-leishmanial compound research, especially since the current chemotherapies available often have debilitating off-target effects. By only focusing on the molecular targets found in *Leishmania*, the potential for mammalian cross-activity greatly decreases. In addition, rodent models for both VL and CL have been used for decades in research settings but have poor translation to human disease; for example, the BALB/c murine model of CL produces measurable lesions but immunologically favors a Th2 response not necessarily mirrored in human disease.

Given the rapid development of drug resistance among pathogens and poor correlation between in vitro drug effects and in vivo efficacy with target-based approaches, research has turned to the potential for multi-target therapies. Importantly, since VL and CL predominantly affect socioeconomically disadvantaged populations, pharmaceutical companies are hesitant to develop and support drug discovery efforts due to the risk of not returning on the investment. By identifying compound classes potent against multiple organisms, larger pharmaceutical companies may be enticed to invest more resources. In addition, CL lesions often become hosts to opportunistic pathogens, like bacterial infections, and HIV-positive patients are more at risk of developing serious symptoms. By utilizing chemotherapies that could combat the parasite and other pathogens, overall costs can be decreased as well as increasing the likelihood of patients fully complying with treatment regimens. However, developing potential multi-target compounds rationally is challenging; target identification is already difficult, and small molecule discovery can be time and resource limiting.

While single-target compounds promise high selectivity, multi-target compounds can help lower the pharmaceutical investment limitation and be effective for comorbidities. There is no one-size-fits-all technique proven to identify the ideal chemical entity to selectively target *Leishmania* with no off-target effects. To truly combat CL incidence and morbidity, a combination of single-target and multi-target compounds need to be further developed and optimized.

## Figures and Tables

**Figure 1 microorganisms-11-02845-f001:**
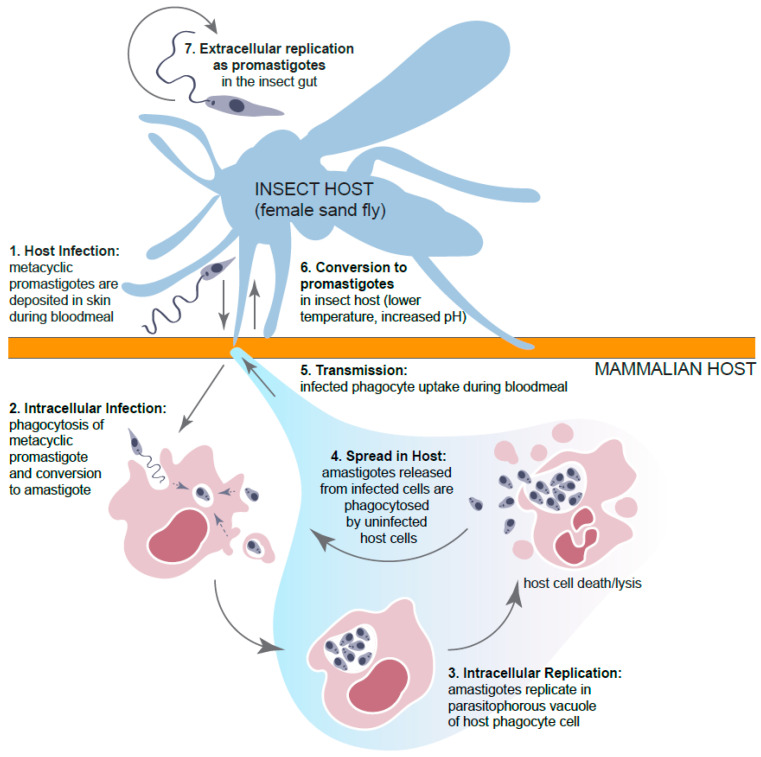
Life cycle of *Leishmania* spp. Promastigotes are deposited into the skin during a bloodmeal of a female sand fly (1). Metacyclic promastigotes are then phagocytosed by immune cells, e.g., macrophages (2). Once in the phagocytic cell, the parasites are isolated into acidic, parasitophorous vacuoles. The acidic environment and increased temperature stimulate the parasite to convert into amastigotes. Multiple amastigotes may reside in a single phagocytic cell, which can result in host cell lysis and death. Amastigotes that are released can be phagocytosed by other phagocytic cells and spread throughout the host (3 and 4). Transmission occurs when phagocytic cells containing amastigotes are taken up during a female sandfly bloodmeal (5). Surviving amastigotes will convert into flagellated promastigotes in the sandfly gut due to lower temperatures and increased pH (6). Promastigotes multiply in the gut and migrate to the sandfly salivary glands (7), and the cycle continues.

**Table 1 microorganisms-11-02845-t001:** Target Product Profile for cutaneous leishmaniasis chemotherapeutics as defined by the DND*i* [31].

	Ideal	Acceptable
Target species	One treatment for all species of *Leishmania*	*L. tropica* or *L. braziliensis*
Safety and tolerability	Well tolerated All adverse reactions (AR)s ≤ grade 1	Safety monitoring at primary health care (PHC) level No major safety concerns Well tolerated in >95% of patients treated Systemic AR ≤ grade 3 in <5% Local AR ≤ grade 2 in <30% No treatment-induced mortality
Contraindications	None	Can be assessed at PHC level
Efficacy	>95% patients with complete clinical cure (100% epithelialization/flattening of lesions at 3 months from treatment onset) Minimal scarring No relapse or development of Leishmaniasis recidivans or mucocutaneous leishmaniasis (MCL) Parasitological endpoint not required	60% epithelialization/flattening of lesion(s) for *L. tropica* and 70% for *L. braziliensis* patients with complete cure Scarring no worse than natural healing <5% rate of relapse or development of Leishmaniasis recidivans or MCL at 1 year
Formulation	Topical or oral	Non-parenteral; few doses if parenteral
Treatment regimen	Topical ≤ 14 days Oral < 7 days	Topical: 28 days Oral: twice daily for 28 days Parenteral ≤ 3 injections
Target population	No restrictions	>9 months of age No efficacy in immunocompromised patients Not for use in pregnancy
Stability	No cold chain At least 3 years at 37 °C	2 years at 4–8 °C
Cost	To be defined	To be defined

DND*i*—Drugs for Neglected Diseases Initiative.

## Data Availability

Data sharing not applicable.

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
