# Peer review of "Drug Discovery for Cutaneous Leishmaniasis: A Review of Developments in the Past 15 Years"

_microorganisms, 2023, doi:10.3390/microorganisms11122845_

Round 1

Reviewer 1 Report

Comments and Suggestions for Authors

The review shows very exciting results on the efforts made to develop drugs to combat cutaneous leishmaniasis (CL) and also presents very interesting conclusions, I enjoyed reading it. I have the following suggestions to improve the review:

Please include references in the second paragraph of the introduction, as the second paragraph contains important information without any references (lines: 38-49)

Please provide more detailed information about the literature search process, details such as criteria for including and/or excluding references, and keywords used to search PubMed. Were Only 5 cutaneous leishmaniasis species used as keywords? (lines 109-114)

Reviewer 2 Report

Comments and Suggestions for Authors

I congratulate the Authors on their work. The manuscript is well written and structured, I have no advice to give to modify or improve the work, which I consider very well performed.

Reviewer 3 Report

Comments and Suggestions for Authors

The review is interesting, it is a fundamental work to reinforce the treatment of Leishmaniasis disease. In addition, it is relevant to emphasize studies of these diseases that are very important and are not given priority to treatment. 

Introduction: I would add about vector control, to reinforce the importance of treatment in patients infected with Leishmania.

Line 35: remove the space after the comma in the bibliography cited throughout the MS.

Line 38-49: supporting literature is missing.

Table 1: italicize strain names: ... lesion(s) for L. tropica and 70% for L. braziliensis...

Line 115: In this section explain the life cycle of the parasite, since we talk about amastigote and proamastigote terms, or clarify these terms for a better understanding. Even a figure could be made explaining the life cycle of the parasite.

Lines 172-186: Several species names should be in italics; even in vivo lack italics.

Lines 215, 245, 268, 285: add the year of the bibliography cited in the text: Alcantar et al. 2020,...

Revise the bibliography according to the journal's norms.

Comments on the Quality of English Language

 Minor editing of English language required
